# A Review of Wood Biomass-Based Fatty Acids and Rosin Acids Use in Polymeric Materials

**DOI:** 10.3390/polym12112706

**Published:** 2020-11-16

**Authors:** Laima Vevere, Anda Fridrihsone, Mikelis Kirpluks, Ugis Cabulis

**Affiliations:** Polymer Department, Latvian State Institute of Wood Chemistry, 27 Dzerbenes Str., LV-1006 Riga, Latvia; anda.fridrihsone@edi.lv (A.F.); mkirpluks@edi.lv (M.K.); cabulis@edi.lv (U.C.)

**Keywords:** crude tall oil, tall oil, fatty acids, rosin acids, polymer materials

## Abstract

In recent decades, vegetable oils as a potential replacement for petrochemical materials have been extensively studied. Tall oil (crude tall oil, distilled tall oil, tall oil fatty acids, and rosin acids) is a good source to be turned into polymeric materials. Unlike vegetable oils, tall oil is considered as lignocellulosic plant biomass waste and is considered to be the second-generation raw material, thus it is not competing with the food and feed chain. The main purpose of this review article is to identify in what kind of polymeric materials wood biomass-based fatty acids and rosin acids have been applied and their impact on the properties.

## 1. Introduction

The success of plastics as a commodity has been significant and polymer materials are a part of everyday life. The advantages of polymers over other materials can be attributed to their adjustable properties, low cost and ease of processing. Worldwide, the manufacturing of polymers grows every year, reaching almost 360 million tonnes in 2018 [1].

In the light of environmental challenges of the 21st century, the development of novel low-cost and scalable monomers from renewable resources is essential. Much attention is being paid to the preparation and application of bio-based polymers [2]. Renewable natural resources are a suitable feedstock of polymer precursors that can be appropriately modified to make new materials with various types of functionalities and adjustable properties [3].

Plant-based oils are considered as a source of bio-based feedstock for material production. However, the use of edible plant oils as a feedstock for polymer production is debatable as edible plant origin oils are used for food and feed production. Tall oil (TO) is a non-edible oil that is derived from renewable woody biomass that is typically cultivated on non-arable land. TO is a side product of the kraft pulping process. Between 1.6 and 2 million tonnes of crude tall oil (CTO) are produced annually [4]. Typically, 30–50 kg TO per tonne of pulp may be recovered from highly resinous species representing about 30–70% recovery [5].

Most investigations regarding TO are dedicated to biofuel production, as well as reduction and cracking reactions [6,7,8,9]. However, if all of TO available in Europe (650,000 tonnes TO per year) was used for biofuel, it would make only 0.2% of the total transportation fuels consumed in 2014 [10]. Tall oil components, i.e., rosin acids (RA) and tall oil fatty acids (TOFA), can be used in several different ways. TO and TO rosin have been applied to improve the hydrophobicity of wood and wood-based products [11,12]. CTO, distilled TO (DTO), and TOFA also can be used for paper impregnating purposes to improve material’s resistance against water [13]. The utilization of different TO components could improve the water resistance of cellulosic fibres [14]. TO and TOFA are mainly used for the production of drying oils, soaps, lubricants, linoleum, paints, and varnishes [15].

Although TO has several more traditional applications, demand for bio-based polymers increases every year. Simple chemical modifications of TO main components, TOFA and RA, can be used in polymer production. This review article is focused on the latest research in this area.

## 2. Tall Oil

### 2.1. Origin of Tall Oil

Wood consists of three main components–cellulose, hemicelluloses and lignin. Along with the main components, there is a group of minor components–extractives. Extractives represent a great diversity of compounds, including phenolic compounds, fatty acids, rosin acids and others. The last two are the main compound groups of CTO. CTO is obtained by delignification process liberating lignin from the wood in cellulose pulp mills [16]. A variety of methods can be used in the delignification process, like mechanical pulping, semi-mechanical pulping, chemical pulping [17].

One of the most widely used chemical pulping methods is alkaline pulping created by C. Watt and H. Burges in the 1850s to extract cellulose fibres from lignin along with fatty acids and rosin acids which was grounded on use of sodium hydroxide solution as a cooking liquor [18]. Alkaline pulping is carried out under conditions at pH above 9. Alkaline pulping is commonly known as Kraft pulping where cooking liquor mainly consists of sodium hydroxide and sodium sulfide, with other sodium salts, such as sodium carbonate and sodium thiosulfate, as minor components for thio-lignin extraction. This cooking liquor is used industrially for delignification [16]. The fats in the form of triglycerides together with rosin acids are dissolved in the cooking liquor. During the saponification step in alkaline conditions, the ester bonds are broken making a black liquor. The black liquor is a complex mixture and contains most of the original cooking inorganic elements and the degraded, dissolved wood substances, namely acetic acid, formic acid, saccharinic acids, other carboxylic acids, dissolved hemicelluloses, methanol, and other components. The black liquor is a sticky, dark brown liquid with unpleasant odour [17].

Afterwards, CTO is isolated from acidified, partially concentrated black liquor via skimming as TO soap has a lower density than the black liquor [17]. The saponified CTO is treated at pH 4 with sulfuric acid at the batch process and diluted sulfuric acid in a continuous process followed by treatment with organic solvent and extraction with water and sodium hydroxide [19]. However, 20–40% of TO soap present in the black liquor may remain soluble and is considered as a waste [19]. Burning of TO soap for energy production is one of the possibilities in the paper production process but it has several disadvantages. In the recovery boiler of the Kraft process, sulfur emissions are increased during the burning process, which leads to decreased boiler efficiency and increased boiler fouling rate, along with process control problems [19]. Therefore, TO is usually recovered from the recovery cycle of the Kraft pulping process to benefit the pulping process.

CTO can be separated in components by distillation as TOFA and RA evaporate at different temperatures. In the distillation phase, there are three continuous separation stages from which 5 products are made. First, CTO is concentrated by evaporation of water and volatile components. Then the process is continued in three distillation columns. The RA is separated from TO in the first column. The remaining mixture is sent to the second and third columns, where TOFA are separated [19].

### 2.2. Composition

The Kraft pulping by-products CTO, DTO, and TOFA change their colour visually from the viscous and dark brown liquid of CTO to the less brown DTO and the yellowish TOFA [13]. CTO is composed of a complicated mixture of RA (30–60%, mainly abietic type acids and pimaric type acids in smaller amounts) (Figure 1), TOFA (30–60%, mainly oleic acid and linoleic acid) (Figure 2) and small amounts of unsaponifiables (5–10%, high-molecular alcohols, sterols and other alkyl hydrocarbon derivatives) [20,21].

Each component of TO has its characteristic impact on tree physiology. The objective of RA consisting mainly of 20-carbon bi- and tricyclic carboxylic acids, is to protect the conifer tree from invading herbivores and pathogens as well as clean and seal the wound from microorganisms [22]. RA are obtained from the resin and contain approximately 90% abietic-type RA, with a tri-ring structure, conjugated double bonds and a carboxylic group with a general formula C_20_H_30_O_2_. The remaining 10% are related to pimaric-type RA with nonconjugated double bonds [23].

Meanwhile, fats in the form of triglycerides provide biological cells of trees with energy. Those fats in the pine trees are occupying the voids called resin canals, go in longitudinal and radial directions and are surrounded by epithelium cells. Mainly these are C-18 straight chain mono- or di-unsaturated fatty acids - oleic and linoleic acids–main components of TOFA. Saturated and tri-unsaturated fatty acids (stearic and linolenic acids) are part of the fats, however in minor amounts.

## 3. The Use of Tall Oil

### 3.1. The Use of Rosin Acids in Polymer Synthesis

The use of RA is welcomed in the synthesis or modification of polymeric materials as it is considered to be an inexpensive, potentially biodegradable and nontoxic raw material [24,25,26,27,28,29]. RA have unique properties that most other natural biomass lack. They are a class of hydrocarbon-rich biomass and can render hydrophobicity to any polymer attached [30,31,32]. RA are composed of hydrophenanthrene structure and contains two chemically reactive centres in the molecule–the carbon-carbon double bonds and the carboxyl group. Therefore, RA can be chemically modified by esterification, isomerization, Diels-Alder addition, amidation reactions (Figure 3) [33,34]. The bulky hydrophenanthrene group present in RA can significantly alter the thermal properties of polymers. Due to conjugated double bonds present in the RA molecules, RA have the property to absorb UV light in the region of 200–400 nm. Therefore, RA is widely used to improve the UV light absorption properties of polymers [14].

### 3.1.1. Reactions of Rosin Acids Carboxylic Group

RA have attracted attention for the modification of polymeric materials, especially natural polymers like starch and cellulose because it is biodegradable and biocompatible. Thermoplastic starch and cellulose materials have several advantages over petroleum-based thermoplastic materials being potentially biodegradable, relatively cheap and available. However, applications of these thermoplastic materials are limited by poor mechanical strength properties and high moisture sensitivity [35,36,37,38]. Modification with RA can help overcome those disadvantages.

Starch can undergo esterification reaction with RA and the reaction proceeds via acid chloride formation (Figure 3B,). The acid chloride formation is applied frequently as the first step in the modification sequence, but reaction conditions are harsh and hazardous. However, the acid chloride formation step can be avoided by applying an enzymatic catalysis system by the use of immobilized lipase catalyst Novozym^®^435 (Figure 3A, Figure 4) [33,39]. Other advantages of the enzymatic esterification reaction are milder reaction conditions and fewer by-products [33]. The esterified starch exhibits higher hydrophobicity and thermal stability while swelling power and solubility in water is decreased and subsequently may find a potential application in waterproof coatings and plastics. However, the transparency of modified starch may diminish [33,39].

RA and FA were used along with ethyl cellulose to formulate bio-based crosslinked elastomers with improved properties. Ethyl cellulose-based thermoplastic elastomers grafted with the RA derivatives and TOFA derivatives (Figure 5) displayed increased thermal stability along with mechanical properties, such as tensile strength, toughness, extensional viscosity, resistance to creep, Young’s moduli and increased hydrophobicity due to rigid structure of RA hydrophenanthrene moiety [40,41]. Moreover, ethyl cellulose-rosin grafted copolymers showed increased UV absorption properties [14] and antibacterial properties [42]. However, too high RA content may contribute to the brittleness of the final material. The effect can be compensated for by the addition of long-chain FA in the polymer system, thus increasing the material’s elasticity [40].

Poly(caprolactone) is linear aliphatic thermoplastic polyester derived from renewable resources with the ability to biodegrade and thus is considered as the replacement of petroleum-based packaging materials. The characteristic properties of poly(caprolactone), like brittleness and high UV light radiation transmission, were overcome by adding RA (Figure 6) [43,44].

Generally, mechanical properties of the polylactic acid matrix, in a similar way as for starch and cellulose materials discussed before, increased with the addition of RA until a certain threshold was reached. After extra RA amounts were introduced in the system unfavourable effects occurred [45]. Tensile strength might slightly decrease with growing RA amount because of the plasticizing effect of RA [44]. Physical properties of RA-substituted polycaprolactone with various molecular weights increased as hydrophobicity improved and glass transition temperature was higher [28,46]. Along with improved mechanical properties, rosin ester–caprolactone grafted copolymers showed a degradability in HCl/THF system within one hour or in phosphate-buffered saline solution within several weeks [28,47].

The chemical modification with RA can be used to improve the properties of silicone rubber. After converting the RA to RA glycidyl ester, the ring-opening reaction is performed using with hydroxy-terminated amino polydimethylsiloxane. The silicone rubber containing RA showed better thermal stability and mechanical properties compared to unmodified silicone rubber due to the strong rigidity and polar hydrogenated phenanthrene ring structure of RA [34].

Modified silicone rubber from acrylpimaric acid-modified aminopropyltriethoxysilane resulted in silicone rubber with higher thermal stability and mechanical properties compared to unmodified silicone rubber which was explained by higher crosslinking density of the modified silicone rubber due to the presence of the rigid hydrogenated phenanthrene ring structure [34]. Similarly, rosin-modified aminopropyl triethoxy silane with comparable properties was prepared [48]. The copolymers of siloxane derivative and ethylene glycol diglycidyl ether modified acrylpimaric acid derived from acrylpimaric acid by esterification reaction with ethylene glycol diglycidyl ether have been reported with improved mechanical and flame retardance properties [49]. The modified rosin containing fluorosilicone rosin with increased polarity was obtained and characterized. It showed increased tearing strength and increased high-temperature thermal stability but diminished oil resistance. Maleated rosin was used in this experiment to produce fluorosilicone rubber with improved quality [50].

Other lesser-known polymers can also benefit from modification with RA. For example, partially replacing styrene with RA monomers in divinylbenzene copolymer, crosslinking and swelling of polymeric materials can be varied. The RA presence increased the decomposition temperature in polymeric monodisperic materials [51]. Hydrogenated rosin used in high adhesion polyurethane acrylate improved the adhesion ability and decreased volume shrinkage during radical UV-curing polymerization process which is a problem with linear analogues [52].

As mentioned before, RA mainly contain abietic acid, which can be used in the synthesis of chiral molecules as abietic acid contains chiral centres in the structure. For example, abietic acid has been used to synthesized chiral polymers with *N*-propargyl abietamide in the presence of rhodium catalyst. The obtained polymer showed enantioselective recognition towards L-alanine, therefore were potentially useful structures for chiral recognition [53]. Poly(*N*,*N*-dimethylaminoethyl methacrylate) copolymer with RA and cationic ammonium moiety in a molecule displayed antibacterial activity against *Escherichia coli* bacteria (*E. coli*) and *Staphylococcus aureus* bacteria (*S. aureus*) due to the hydrophobicity and fused-ring structure of the RA [54]. In the foundation of these antibacterial properties were amphiphatic properties of the copolymer, which was partially charged and partially uncharged and had cationic hydrophilic group and bulky hydrophobic hydrophenantrene group. Therefore, the antimicrobial properties of polymeric materials were improved by the incorporation of RA into matrix [45].

### 3.1.2. Modifications Starting with Diels-Alder Addition

Levopimaric acid, the most abundant RA, obtained by thermal treatment of abietic acid (Figure 3C) can undergo Diels-Alder addition reactions, which are widely used in rosin modification reactions. Acrylic acid, fumaric acid or maleic anhydride are often used as starting materials in the reaction with levopimaric acid in Diels–Alder reactions [41].

Polystyrene-rosin microspheres can be prepared by suspension polymerization starting with RA Diels-Alder reaction with acrylic acid. Acrylic rosin ester is then used instead of commonly used divinylbenzene for copolymerization reaction with styrene. Monodispersity and particle size was affected by styrene/acrylic rosin ester ratio [55]. Another reaction adduct between levopimaric acid and acrylic acid bearing two carboxyl groups in a molecule can be used in polycondensation or direct polymerization reaction to produce polyesters with good thermal stability and high dielectric properties [56].

Maleopimaric acid polyester polyol–three reactive carbonyl groups containing molecule–was prepared from maleopimaric acid (Figure 7) and used further in polyurethane emulsion production. RA containing polyol addition increased particle size of the polyurethane dispersion and tensile strength but decreased elongation at the break because of lowered flexibility of molecular chains and elevated crosslink density at increasing rosin moiety ratio [57]. A nanoscale polyurethane microsphere monodispersion from rosin acrylate with potential application in the adsorption field showed good thermal stabilities and average particle size of 120 nm [58].

The use of RA in epoxy resins has been studied. The most used monomer for the production of epoxy resins was bisphenol A. The curing agents–maleopimaric acid and maleopimaric acid imidoamine–have been used to make epoxy resin with diglycidyl ether of bisphenol A. The curing agents displayed good curing abilities along with high thermal stabilities, especially when maleopimaric acid imidoamine was used [59].

Bisphenol A is known to have a negative impact on animal health like alterations in brain chemistry and structure, behaviour, the immune system, enzyme activity, the male and female reproductive system in a variety of animals [60] therefore scientists are searching for substitutes for bisphenol A. The chemical transformations of RA allowed generating a two-component fully rosin-based epoxy system from epoxy resin and crosslinker–both consisting of modified RA derivatives. Epoxy resin was synthesized from dipimaryl ketone obtained from isomerized abietic acid (Figure 3C). After Diels–Alder cycloaddition of maleic anhydride and epoxidation tetraglycidyl dimaleodipimaryl ketone was obtained as epoxy resin (Figure 8). Dirosin-maleic anhidryde imidodicarboxylic acid was synthesized from RA and 1,1-(methylendi-4,1-phenylene)bismalimide in Diels-Alder cycloaddition reaction and was used as a crosslinker. Thermal stability of cured resin consisting of a fully bio-based epoxy system from rosin containing epoxy resin and rosin containing crosslinker was somewhat higher than that of commercial epoxy resins with bisphenol A [30]. Rosins with their rigid structures gave properties similar to those of bisphenol A and showed improved the resistance to water [61,62].

However, epoxidized rosins tend to be solid at room temperature which is undesirable for their processing into cured materials [61]. Amine-type curing agents are widely used for epoxy resins as they are extremely versatile and can be mixed with epoxy resins at any ratio and epoxides cured with amine curing agents exhibit excellent chemical resistance and mechanical properties [61]. Addition of 2-ethyl-4-methylimidazole as catalyst greatly decreased the curing temperature and promoted the completion of cure reactions between eugenol epoxy and the rosin-derived maleopimaric acid synthesized from abietic acid and maleic anhydride. Imidazole-type catalysts are often used for high-temperature curing agent, anhydride–to reduce the curing temperature [63]. Another option to decrease the melting point of rosin component of epoxy resin can proceed by epoxidation of double bonds of the rosin oligomer (polygral–a mixture of abietic acid and its isomers, dimers and trimers) to obtain the exocyclic thermoset with amino curing agent without bisphenol A containing substrate [61]. Also, more flexible fragments may be included into a molecule of modified RA with fumaric acid to fumaropimaric acid by Diels–Alder reaction pathway to make epoxy resin more suitable with better mechanical properties as RA tend to make cured resin epoxide brittle due to rigid structure of the RA [62].

Polybenzoxazines with high thermal stability were synthesized from RA. In step one, the rosin is converted with maleic acid to maleopimaric acid with the following transformation with 4-amionobenzoic phenol to maleopimaric acid imidophenol. In the last step, cyclization reaction with aniline and paraformaldehyde or 4-aminobenzoic acid and paraformaldehyde was achieved. Each of the products can then be thermally polymerized at different temperatures. The thermal stability of the rosin containing products was better than that of commercial polybenzoxazine product based on phenol, paraformaldehyde, and anyline due to the steric hindrance of the hydrogenated phenanthrene ring [64].

### 3.2. The Use of Tall Oil Fatty Acids

Nowadays TOFA are regarded as polymer precursors and have gained increasing interest in different spheres of the academic and industrial areas. It is a good source for the production of different polymers, such as polyesters, polyethers, polyurethanes, polyamides, epoxy resins, alkyd resins [2,65,66,67]. Vegetable oils containing fatty acids are regarded as a starting material for polymer synthesis, however, TOFA have several advantages in comparison with other oils studied for polymer synthesis. TOFA, which are one of the main components of TO, are free of glyceride backbone in the molecules which were eliminated during the acidification process in Kraft pulping of cellulose liberating procedure to yield free TOFA. Another important advantage of TOFA is the sufficiently high iodine value (around 155 g of I_2_/100 g) compared to the vast majority of vegetable oils (for example, for soybean oil, the iodine value is 100–170 g of I_2_/100 g [68,69], and for rapeseed oil less than 125 g of I_2_/100 g [69,70]). The higher iodine value means that TOFA have a larger amount of double bonds present in their structure available for chemical reactions [71].

The use of TOFA as a renewable feedstock for polymer synthesis can be achieved through chemical modification [26,71,72,73,74,75,76,77,78,79,80,81]. TOFA have two reactive sites in the molecule–a carboxylic group and one to three double bonds [26]. That allows using TOFA for both esterification and epoxidation reactions (Figure 9).

#### 3.2.1. Tall Oil-Based Polyol Development

##### Carboxyl Group Reaction in Tall Oil Fatty Acids

Polyester or polyamide precursors can be produced by different chemical modification of TOFA. Yakushin et al., 2016 synthesized TOFA based polyols using reactions that yielded diethanolamide (obtained in TOFA amidation reaction with diethanolamine) and triethanolamine esters (obtained in TOFA esterification reaction with triethanolamine), respectively. TOFA diethanolamine and triethanolamine ester polyols were synthesized in a relatively simple one-step reaction. However, the synthesis of TOFA triethanolamine polyol required higher reaction temperature of 180 ± 5 °C and larger amine/TOFA ratio. Synthesized TOFA polyols were successfully applied in bio-based polyurethane film production [74].

TOFA were esterified with trimethylolpropane without a catalyst at different molar ratios of trimethylolpropane to TOFA. These polyols proved to be less active than the nitrogen atom containing diethanolamine and triethanolamine [82].

Frust et al. used linoleic acid as a reference substance in methoxycarbonylation reaction with in situ prepared palladium catalyst. Linoleic acid was functionalized with CO in methanol adding another carboxyl group in the form of methyl ester and one extra carbon atom to the structure (Figure 10). Other by-products of the reaction were isolated and identified being a mixture of isomers of keto esters with the keto group at different positions along the chain, a mixture of methyl methoxyoctadecanoates, and a mixture of triesters. If TOFA, which are a mixture of mainly oleic acid and linoleic acid, are used, a mixture of saturated and unsaturated methyl esters is obtained. And the proportion of saturated and unsaturated methyl esters in the mixtures can be varied at different reaction conditions [83].

##### Reactions at Tall Oil Fatty Acids Double Bond

Epoxides obtained from TOFA may be converted to alcohols, glycols, carbonyls, alkanolamines, substituted olefins, polyesters, polyurethanes and epoxy resins [84]. Nowadays epoxidized oils commercially are produced via Prilezhaev epoxidation process (Figure 11) [85]. In the Prilezhaev epoxidation, a peroxy acid is prepared by an in situ reaction of an acid with hydrogen peroxide [86,87,88]. Few drawbacks of Prilezhaev reaction include the production of highly corrosive waste, undesirable ring-opening reactions and polymerization of the product under acidic conditions [89]. To overcome these obstacles ion exchange resin Amberlite^®^ IR-120 was used by Vanags et al., 2018 for epoxidation of TOFA followed by polyol formation between epoxidized TOFA and triethanolamine in the presence of heterogeneous catalyst–acidic clay Montmorillonite K10 for oxirane ring-opening [90].

Kirpluks et al., 2019 used immobilized lipase catalyst Novozym^®^435 for double bonds epoxidation present in TOFA. The use of enzyme allowed the epoxidation reaction to be carried out at mild conditions (40 °C) and without any use of organic solvent. At the same time, the product with relatively high epoxide oxygen content was achieved (4.49–6.00%) with the highest unsaturated bond conversion of 72.3% [91]. Hydrogen peroxide has a significant effect on the reaction rate and degree of epoxidation in this type of reaction [91,92], as well as the mode of stirring [91].

##### 3.2.2. The Use of Tall Oil and Tall Oil Fatty Acid-Based Polyols in Polyurethane Formation

The formation of polyurethanes proceeds via reaction of isocyanate containing two or more isocyanate groups and polyol containing two or more hydroxyl groups, resulting in the formation of linear, branched, or cross-linked polymers [25].

TO based polyols have been studied for their use in rigid polyurethane foam production. Extensive work in this field has been carried out by Latvian State Institute of Wood Chemistry [73,74,76,77,78,90,91,93]. Cabulis et al., 2012 produced rigid polyurethane foams with a density of 45 ± 5 kg/m^3^ from TO (with RA content 20%) polyol obtained in the amidation process. The maximum content of the TO in ready foams reached 26% [75].

In polyurethane materials, soft segments are built of polyol (long-chain diols), and the glass transition temperature of the polyols is much lower than room temperature. Hard segments are derived from diisocyanate and short-chain diols [94]. Thermal stability of the polyurethane material is influenced by changing the ratio of the hard segments to soft segments [95]. Polyurethane films were obtained from tall oil amide and ester type polyols that contained a different amount of RA. The effect of RA content in TO on the characteristics, such as density, glass transition temperature, thermal stability, mechanical properties and adhesive strength, of the obtained polyurethanes was investigated [74]. The properties of the obtained polyurethanes were greatly influenced by the TO polyol type, as well as RA content in the synthesized TO polyols. Polyurethane amides obtained in diisocyanate reaction with TOFA diethanolamides showed increased glass transition temperatures due to hard urethane-amide segments and tensile strength along with modulus of elasticity [74,77]. Polyurethane amides produced from TO demonstrated higher thermal resistance and rigidity but lower tensile strength than the polymers produced with the use of petrochemical-based ethylene glycol [77].

In polyester urethane macromolecules the fatty acid group is attached to nitrogen atom via ethylene chain that makes the molecule more flexible [76]. These macromolecules are synthesized from triethanolamine esters and TO. Glass transition temperature of polyester urethanes was lower than that of the poly(urethane amides) due to the dangling chains of triethanolamine residue, therefore weaker intermolecular interaction. The polyol here acted as a plasticizer. The elongation at break was nearly twice as big as that for the polyurethane amides. The shear bond strength and thermal stability were also increased for triethanolamine TO polyol containing polyester urethanes. Regarding RA content in the polyurethane materials, RA increased the density of the material due to the rigid phenanthrene structure of RA and also influenced the tensile strength and modulus of elasticity but RA may decrease the adhesion [74]. Rigid polyurethane foams produced from TO polyol with triethanolamine with a density below 50 kg/m^3^ and at isocyanate index below 175 showed water absorption below 2 vol.% during 28 days. The designed material with these characteristics can be used as a buoyancy material in boat construction, shipbuilding and production of life-saving equipment [73].

Volatile organic solvent-free polyurethane coatings were prepared based on the synthesized polyols between TOFA and trimethylolpropane without the use of solvent [82]. The obtained coating containing trimethylolpropane and TOFA polyol proved to have higher tensile strength, modulus of elasticity with higher elongation at break than the previously mentioned polyurethane materials from diethanolamide or triethanolamine esters of TOFA. The initial temperature of decomposition was also higher [82].

Polyurethane foam produced from bio-based polyols showed higher flammability compared to those produced from synthetic materials [95]. The polyurethane foams with increased apparent densities were prepared from TO diethanolamide polyol by incorporation into the system paper production waste sludge (consisting of calcium carbonate, kaolinite, dolomite, silica, etc.) as a flame retardant. The obtained material with smaller cell structures suppressed the flame and flame spread due to the release of H_2_O and CO_2_ [95]. More classical flame retardants were tested on polyurethane coating prepared with TO triethylamine ester and a retardant system consisting of ammonium polyphosphate/pentaerythritol/melamine (3:1:1) showed the best flame-retardant properties [93].

##### 3.2.3. Tall Oil Fatty Acid Use for Alkyd Resins and Anticorrosive Smart Microcapsules Based Self-Healing Agents

Oil-based resins, such as alkyds, epoxies, and polyesteramide, have significantly proven their applicability as a binder in surface coatings [96]. Alkyd resins are widely used in the coating industry [97], and they are the most versatile binder used in architectural, industrial, and decorative coatings [98]. Alkyd resins are polyesters that are synthesized by polycondensation reaction of dibasic acid or acid anhydride, polyol with two or more hydroxyl groups and oil or oil-derived FA [99,100] where TOFA can serve as a source of fatty acids.

Rämänen et al. synthesized alkyd-acrylic copolymers from TOFA, pentaerythritol and isophtalic acid for application as a paper/paperboard coating. The author found that, with increasing TOFA content in the alkyd resin, the copolymer film coating showed higher brittleness, influenced water repellency and decreased oxygen permeability because of crosslinking formed during the drying of the copolymer coating caused by the double bonds in the fatty acid chains [101].

Hyperbranched polymers—the non-linear type of polymers with highly branched structure and a large number of end groups—have found their use in coatings. The chemical structure of the polymers and the type of end groups determine the rheological properties, cross-linking abilities, chemical resistance, and mechanical properties of these coatings [102]. Alkyd resin with hyperbranched core and TOFA attached to the hydroxyl groups by esterification reaction of the central part of the molecule was prepared by Murillo et al., 2010 [80]. Alkyd resins showed excellent adhesion, flexibility, drying time, gloss and chemical resistance. These polymers have higher elasticity, lower viscosity and shorter drying time compared to conventional alkyd resins and they make nanometric and polydisperse system. Hyperbranched polyester was formed with hydrophobic TOFA molecule parts and reactive double bounds toward the outside of the molecule and hydrophobic part located in the middle part. This core part of the hyperbranched alkyd resin molecule was made by bulk polymerization from 2,2-bis(hydroxymethyl)propanoic acid with trimethylolpropane, pentaerythritol and ethoxylated pentaerythritol [83]. TOFA were also used in the synthesis of styrene-hydroxyethyl acrylate copolymer-based silicone-alkyd resins with comb-type structural morphology, low branching degree, a high number of terminal chains and compact architecture generating low viscosity. Styrene-hydroxyethyl acrylate copolymer alkyd resin can be synthesized with comb-type structural morphology by incorporation of macromonomer consisting of dimethylol propionic acid and TOFA. These macromonomers had a plasticizing effect due to TOFA presence in the macromolecule [81].

TOFA was also used to form anticorrosive smart microcapsules based self-healing agents. In the foundation of the healing process is the ability of the microcapsules to form a new layer through crosslinking reaction of the healing agent with oxygen, moisture, sunlight or other crosslinkers by way of the release of the healing agent. Zhang et al. reported the development of fatty acid-based epoxy ester as a novel healing agent in microcapsules for self-healing coatings. A commercially available TOFA-based epoxy ester resin (70 wt.% resin and 30 wt.% xylene) was used in the study. Epoxy ester synthesized from epoxy resins and TOFA as the healing agent for self-healing coating was cured at room temperature through auto-oxidation without additional crosslinkers by free radical mechanism initiated from unsaturated sites of TOFA moiety. The synthesis of microcapsules containing epoxy ester resin started with urea-formaldehyde prepolymer formation in alkaline conditions with following encapsulation of epoxy ester of TOFA under acidic conditions in water emulsion [24]. Rämänen, P. & Maunu reviewed the advantage of the NMR spectroscopy methods in the structure of TOFA-based alkyd resins and alkyd–acrylic copolymers before and after oxidative drying process to describe film formation and film properties [103].

## 4. Conclusions

This review study encapsulates an effort to conduct a general overview on the use of wood biomass-based fatty acids and rosin acids use in polymeric materials, covering topics from origin and composition of tall oil to the use of tall oil components in different polymeric materials.

Past years have been productive in research in the field of tall oil and its products for novel polymeric material synthesis methods. Rosin acids mostly have been used as an additive in polymeric materials. Rosin acids were used to improve properties of biopolymers such as starch, cellulose and caprolactones, as well as epoxy resins and silicone rubber. Biocompatibility and antimicrobial properties are the most important properties of rosin acids. Unfortunately, large quantities of rosin acids can lead to a more brittle and less flexible final product.

Tall oil fatty acids are used as a major component in polyurethane production. In general, bio-based polyurethanes have been prepared by first converting tall oil fatty acids into polyols. Multiple methods have been used to synthesize polyols. Tall oil fatty acids can reach more than 25% in ready polyurethane foams. Moreover, another major application for tall oil fatty acids is alkyd resin production, wherein they are used as a source of chain extenders. The addition of tall oil fatty acids can improve the mechanical and thermal properties of alkyd resins and also polyurethanes.

## Figures and Tables

**Figure 1 polymers-12-02706-f001:**
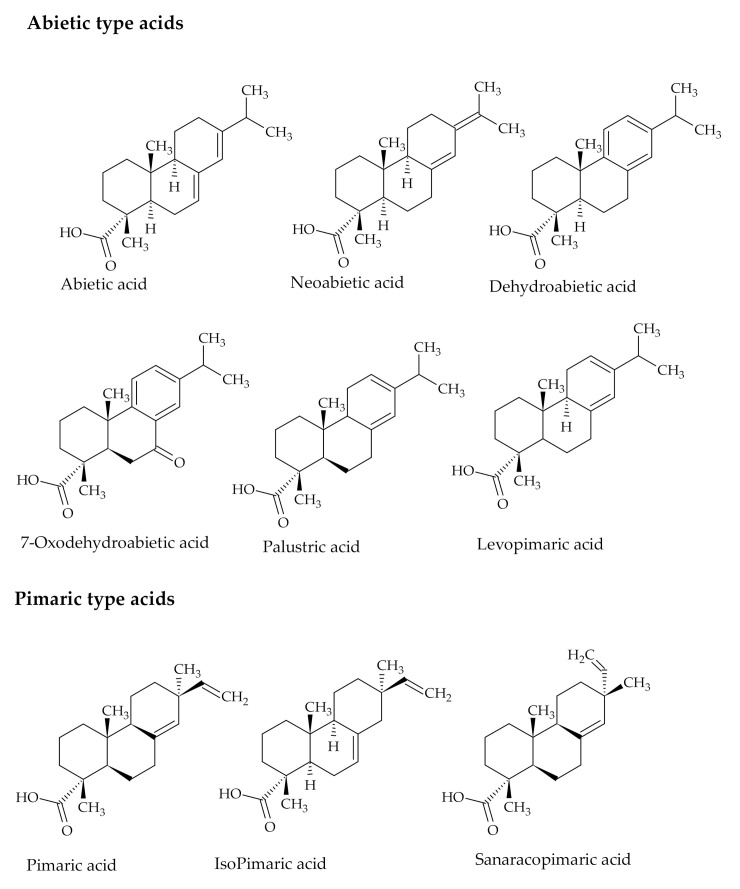
The chemical structures of RA.

**Figure 2 polymers-12-02706-f002:**
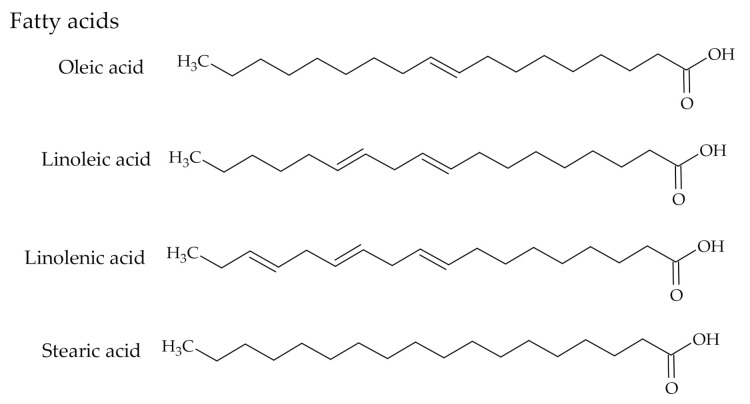
The chemical structures of TOFA.

**Figure 3 polymers-12-02706-f003:**
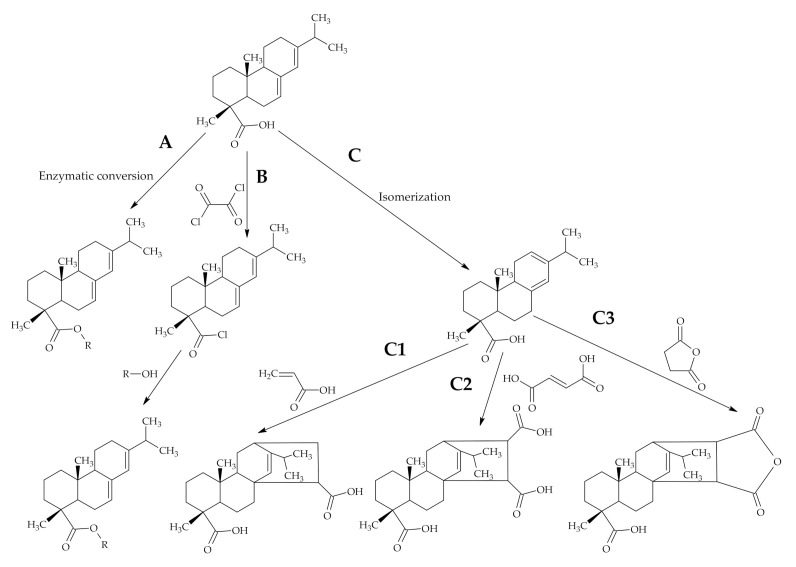
Basic abietic acid transformations to be applied in polymerization reactions, where (**A**)—enzymatic conversion; (**B**)—acid chloride formation following esterification; (**C**)—isomerization different Diels-Alder cycloaddition.

**Figure 4 polymers-12-02706-f004:**
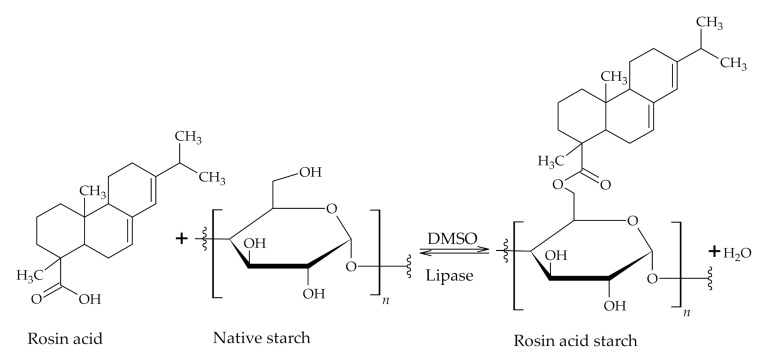
Esterified starch with RA catalyzed by lipase.

**Figure 5 polymers-12-02706-f005:**
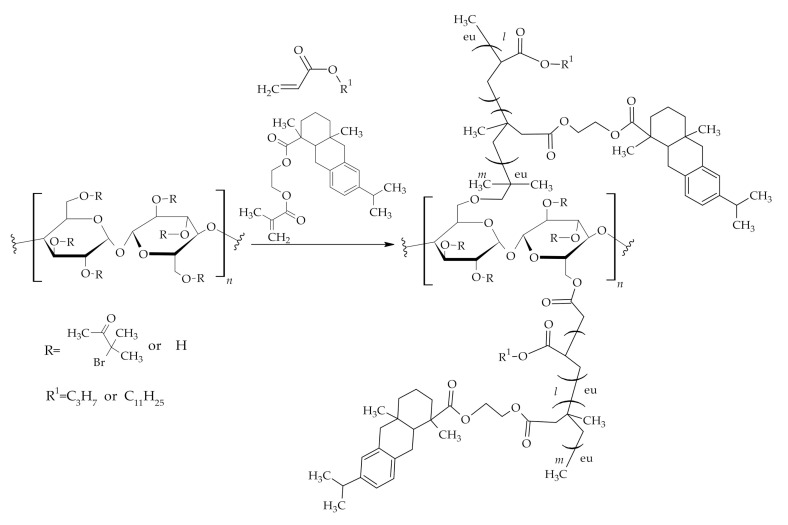
Cellulose modification with RA and TOFA.

**Figure 6 polymers-12-02706-f006:**
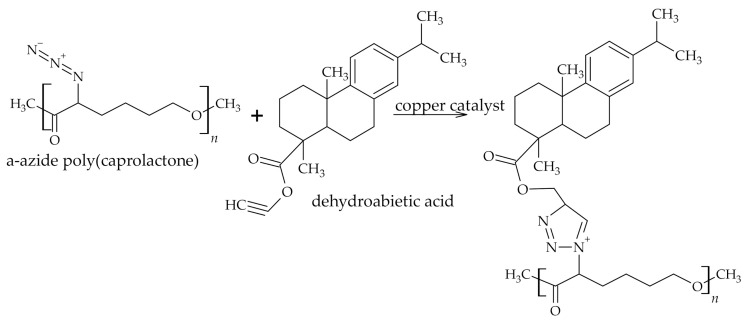
RA modified polycaprolactone.

**Figure 7 polymers-12-02706-f007:**
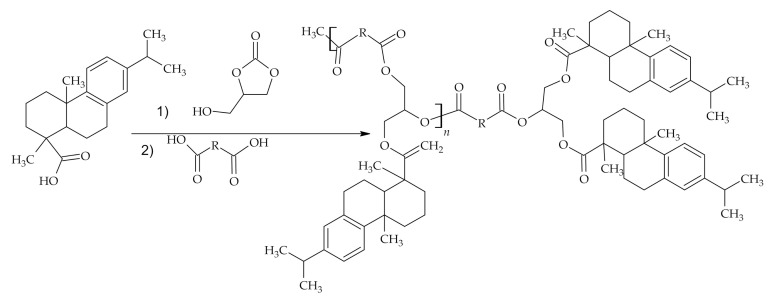
Schematic view of maleopimaric acid polyester polyol synthesis.

**Figure 8 polymers-12-02706-f008:**
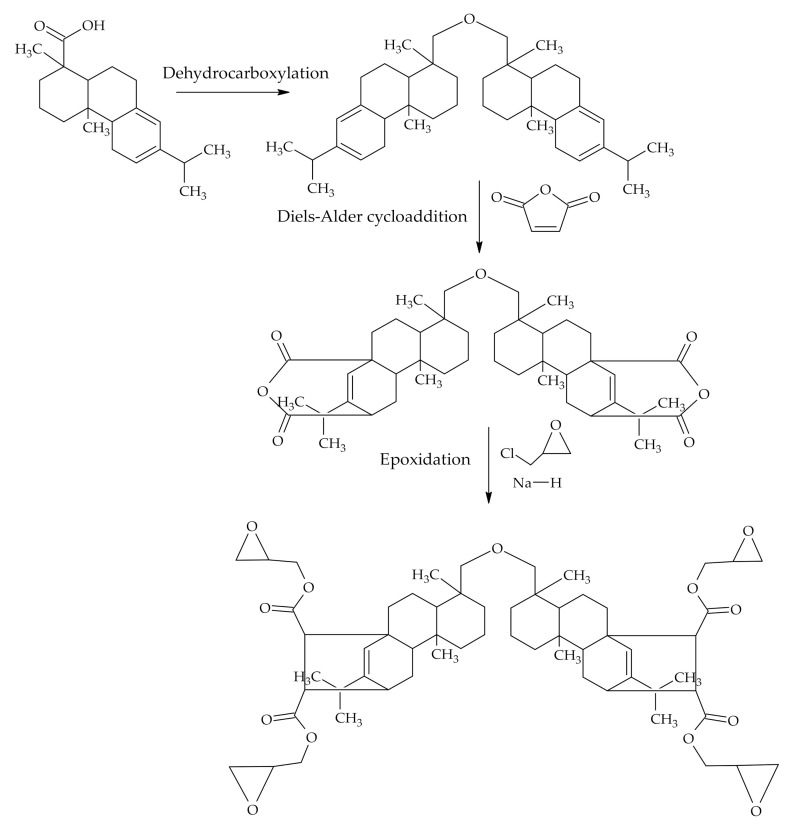
Epoxidation of levopimaric acid.

**Figure 9 polymers-12-02706-f009:**
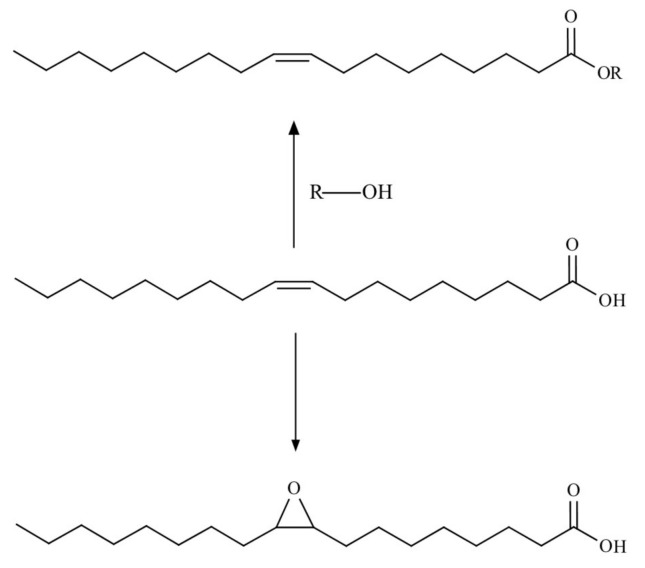
Basic TOFA transformation.

**Figure 10 polymers-12-02706-f010:**
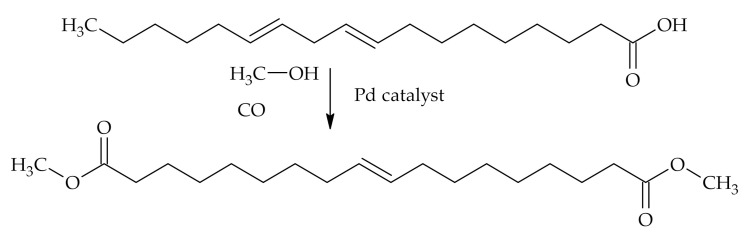
Methoxycarbonylation of linoleic acid.

**Figure 11 polymers-12-02706-f011:**
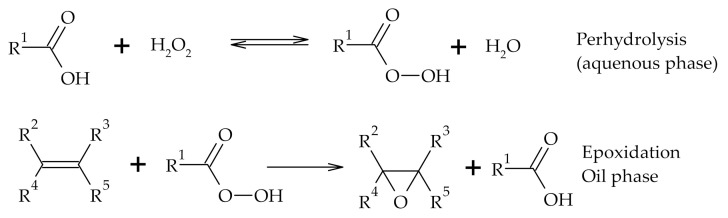
Reactions involved in epoxidation with Prilezhaev method.

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
