# Peer review of "A Review of Wood Biomass-Based Fatty Acids and Rosin Acids Use in Polymeric Materials"

_polymers, 2020, doi:10.3390/polym12112706_

Round 1

Reviewer 1 Report

I think this manuscript is not accepted for publishing. The comments are as in following: 1. I did not find you list any track of wood biomass in this review 2. The amounts of the table is not enough to support a review.

Author Response

Thank you very much for your comments.

Reviewer 2 Report

In the present review the authors exhaustively reported the use of RA and TOFA in the field of polymers, as additives or precursors for their synthesis. The manuscript was correctly organized and no objection can be found in the content, that is approved in its present form.

I have only a comment regarding the Tables that, in my opinion should be removed (in such form, they do not give any further information to the reader) and a suggestion to replace them with tables summarizing the written content and/or explicative figures on detected enhancements (thermal stability, as an example...)

Author Response

Dear Reviewer,

Thank you for great and constructive review.

We have taken into account Your feedback 2 and we have removed the Tables. However, we believe that it is best not to add any new tables with explicative figures on detected enhancements. This is due to the fact that most often in the papers we have reviewed the detected enhancements are not reported in exact numbers/ percent’s not rather there have been a textual description that a specific enhancement has been observed. Thus, we believe that the table mentioned before would be insufficient and not delivering much to the manuscript.

Moreover, we have given English one more time for additional improvements and have replaced Figure 5 due to copyright issues.

Thank you.

Laima Vevere, MSc.

Researcher

Polymer Department

Latvian State Institute of Wood Chemistry

27 Dzerbenes str., Riga, Lv1006

Reviewer 3 Report

It is undeniable that the authors have done a good work by reviewing the publications on the wood derived fatty and rosin acids and their use in polymeric products. The concept is good, the review is equilibrated and the bibliographic references - 103- are fair for a short review.

The authors show the main derivatives composition extracted from the wood: crude tall oil, tall oil and tall oil fatty acids describing also the problematic of the composition of these derivative, especially that of TOFA. Their use in the synthesis of polymers in well represented by clear illustrations of starting and produced chemical structures. Generally, the chemical reactions of carboxylic functions of TOFA are given and explained by clear and completed schemes. The obtained polymers are discussed in terms of properties and uses.

In my opinion, is a good and correct review, very useful for the domain. So I recommend its publication.

Author Response

Thank you very much for your comments.